# Prenatal exposure to the probiotic *Lactococcus lactis* decreases anxiety-like behavior and modulates cortical cytoarchitecture in a sex specific manner.

Natalia Surzenko[1,2◉]*, Eneda Pjetri[1◉], Carolyn A. Munson[1], Walter B. Friday[1], Jonas Hauser[3], Ellen Siobhan Mitchell[3]*

**1** Nutrition Research Institute, University of North Carolina at Chapel Hill, Kannapolis, NC, United States of America, **2** Department of Nutrition, Gillings School of Global Public Health, University of North Carolina at Chapel Hill, Chapel Hill, NC, United States of America, **3** Société des Produits Nestlé S.A., Nestlé Research, Lausanne, Switzerland

◉ These authors contributed equally to this work.
* surzenko@email.unc.edu (NS); esiobhanmitchell@gmail.com (ESM)

**Data Availability Statement:** All relevant data are within the paper and supporting information file. Each data file can also be access through

## Abstract

Development of the cerebral cortex may be influenced by the composition of the maternal gut microbiota. To test this possibility, we administered probiotic *Lactococcus lactis* in drinking water to mouse dams from day 10.5 of gestation until pups reached postnatal day 1 (P1). Pups were assessed in a battery of behavioral tests starting at 10 weeks old. We found that females, but not males, exposed to probiotic during prenatal development spent more time in the center of the open field and displayed decreased freezing time in cue associated learning, compared to controls. Furthermore, we found that probiotic exposure changed the density of cortical neurons and increased the density of blood vessels in the cortical plate of P1 pups. Sex-specific differences were observed in the number of mitotic neural progenitor cells, which were increased in probiotic exposed female pups. In addition, we found that probiotic treatment in the latter half of pregnancy significantly increased plasma oxytocin levels in mouse dams, but not in the offspring. These results suggest that exposure of naïve, unstressed dams to probiotic may exert sex-specific long-term effects on cortical development and anxiety related behavior in the offspring.

## Introduction

Neocortical development is influenced by a variety of environmental factors, including maternal nutrition, stress and exposure to pathogens. In terms of cortical patterning and growth, the impact of essential micronutrients, such as choline, has been well-characterized; however, less is known on the influence of food-derived microbiota, e.g. probiotics and alterations in cortical development at different stages [1]. While the fetal gut is sterile before birth, it is increasingly clear that factors from the maternal microbiota can induce specific patterns of gene expression

researchgate with the following DOIs: DOI: 10.13140/RG.2.2.32607.48808 DOI: 10.13140/RG.2.2.25896.60163 DOI: 10.13140/RG.2.2.25896.60163 DOI: 10.13140/RG.2.2.29252.04480 DOI: 10.13140/RG.2.2.19185.71527 DOI: 10.13140/RG.2.2.23694.59206

**Funding:** NS and EP received funding from Société des Produits Nestlé SA. The funders had no role in study design, data collection and analysis, decision to publish, or preparation of the manuscript. URL: www.nestle.com

**Competing interests:** I have read the journal's policy and the authors of this manuscript have the following competing interests: ESM and JH are or were employees of Société des Produits Nestlé SA, a for-profit institution, and received support in the form of salaries. This does not alter our adherence to PLOS ONE policies on sharing data and materials.

in the fetal gut, as well as in the brain [1–3]. After birth, microbial colonization of the infant gastrointestinal tract has been shown to have widespread influence on brain development and behavior [4]. For example, mice whose gut microbiota were depleted via antibiotics or who were raised in a germ-free facility, demonstrated exaggerated responses to stress and social stimuli, and displayed neurochemical and brain structural abnormalities [5, 6]. Recently, neonatal exposure to probiotics, such as *Bifidobacterium longum* and *Lactobacillus rhamnosous*, have been shown to reverse maladaptive learning behaviors in innately anxious mice [7]. Probiotics have also alleviated behavioral effects in other models of anxiety disorders, such as those induced by maternal separation, social defeat or other types of early life stress [8–10]. Mounting data on brain-gut axis has revealed several pathways where probiotic treatment can affect microbiota populations, brain signaling, and subsequently anxiety in mouse models of disease [1, 7, 9]. However, there is less consensus on the behavioral effects of probiotics in naïve, wild type mice lacking exposure to stress paradigms.

Neurochemical investigations have revealed that probiotic treatment reduced inflammatory cytokines and stress-related hormones, which are often chronically activated in anxiety disorders [7, 11]. For instance, modulation of the vagus nerve has been implicated in behavioral effects of probiotics, since vagal resection blocks the anti-inflammatory, anxiolytic activity of probiotics [11]. Additionally, probiotics may impact brain development by improving maternal health and immune-mediated stress responses. Exposure to probiotics, such as *Lactobacillus reuteri*, can increase plasma oxytocin [12], potentially enhancing nurture instincts and pup handling in treated dams. Studies such as these implicate changes in neuroendocrine signaling brought on by plasma metabolites derived from the probiotics or from specific commensal microbiota interacting with the probiotics. In fetuses with rudimentary microbiomes, metabolites from maternal microbiome access the fetal blood stream via placental transfer [2]. However, there is a dearth of knowledge on how probiotic exposure in utero, i.e. before postnatal colonization, affects structural brain development in wild type, naïve neonates. In the present study, we investigated whether neonatal exposure to the probiotic *Lactococcus lactis* (*L. lactis*) induces long-lasting changes in cortical layer structure. We found that the density of distinct types of cortical neurons were changed by probiotic exposure, as well as increased proliferation of cortical neural progenitor cells. Additionally, we measured anxiety and emotional learning capacity in probiotic-exposed mice and found that maternal L. lactis exposure ameliorates anxiety-related behavior in offspring.

## Materials and methods

### Animals

All experiments were performed at the David H. Murdock Research Institute Center for Laboratory Animal Science facilities in accordance with the standards of the U.S. National Institutes of Health Guide for Care and Use of Laboratory Animals and were approved by the Institutional Animal Care and Use Committee at this facility. *Nestin-CFPnuc* transgenic mice were generously provided by Grigori Enikolopov (Cold Spring Harbor Laboratory, Cold Spring Harbor, NY, USA) [13] and were maintained on a mixed C57BL/6J and C57BL6/N background (97% C57BL/6J). Mice were habituated to modified AIN93G diet (#103186, Dyets Inc., Bethlehem, PA) for at least two weeks before breeding and until the end of the experiments.

Pregnant dams were randomly divided in two groups: one group was given the probiotic *Lactococcus lactis* (*L. lactis*) in drinking water (5X10^5 CFU/ml) from gestational day (GD) 10.5 to postnatal day 1 while the other group (Control) received drinking water only. The drinking water was refreshed daily. Probiotic CFU was confirmed to remain unchanged after study completion.

## Behavioral tests

Mice were tested at 10–13 weeks in the behavioral battery for locomotor activity and anxiety-like behavior in open field, exploratory behavior in object investigation, anxiety-like behavior in the light-dark box test and learning and memory in fear conditioning test [14]. In the control group, nine male and ten female mice were tested, while in the *L.lactis* exposed group, ten males and twelve females were tested. The mice were littermates from multiple litters. The mice were kept on a 12:12 hr light-dark schedule and were tested in the light phase, in addition, male and female mice were tested on separate days and were given at least 48 hr between the tests.

**Open field test.** A widely used test to assess locomotor activity and as a measure of anxiety-like behavior in a novel, stressful environment [15]. The open field was a circular arena with diameter 80 cm and height 35 cm that was evenly illuminated with white light. The mice were allowed to explore for 5 min while the test was recorded and analyzed with an automated tracking system, Ethovision XT® (Noldus Information Technologies, The Netherlands). We measured total distance moved, velocity, latency to center zone, and total duration in the center zone.

**Object investigation test.** In this test, we measured the exploratory activity of a mouse towards a novel non-aversive object placed in the center of the arena after they are familiar with the environment [16, 17]. The mice started facing the wall in the same open field arena and allowed to habituate for 2 min. Then, the novel object is introduced in the center of the arena and the mice are allowed to explore it freely for 3 min. The total time investigating the object was scored manually.

**Light-dark box.** The light-dark exploration test is used to measure anxiety-like behavior and the conflict between rodents' exploratory behavior and aversion to open and brightly illuminated areas [18].

The test box ($44 \times 21 \times 21$ cm; l x w x h) is divided into two unequal compartments by a partition (1 cm) with a small aperture ($5 \times 7$ cm) in the center by the floor level, allowing the mice to move freely between the chambers. The dark chamber, 1/3 of the box, is black and covered with a lid. The light chamber is white and brightly illuminated. The test mouse was placed into the dark chamber facing the end wall and allowed to explore for 5-min. Ethovision XT® was used to record locomotor activity in the light zone. Transition between chambers, when all four paws are placed in one chamber, was scored manually. Duration in each chamber and the latency to go to the light chamber was also recorded.

**Contextual and cued fear conditioning test.** Contextual and cued fear conditioning was performed using a conditioned fear paradigm over 3-days using Near-Infrared image tracking system (MED Associates, Burlington, VT) [19]. On the first day, after an initial 2-min exploration time (background activity), mice were exposed to a 30-s tone (85 dB, 2800Hz), followed by a 2-s scrambled foot shock (0.75mA) (CS1) under white light conditions. Mice received 3 additional tone-shock pairings (CS2 and CS3), with 80-s between the stimuli pairings, totaling a 9.2-min session. The response to the shock was measured with automatic assessment of the levels of freezing (immobility, except for breathing) using the Video Freeze (MED Associates Inc.) software. All mice learned the association between the tone and the shock, and the response to the tone increased with each pairing.

## Immunohistochemistry

P1 brains, representing pups from 4–5 independent litters, were fixed in 4% PFA, cryopreserved through incubation in 10%-20%-30% sucrose/1 X PBS gradient over 72 hours at 4˚C, mounted in O.C.T. compound and stored at -20˚C. Brains were cryo-sectioned coronally at

20 µm over a series of 7 slides, such that each slide contained representative non-consecutive brain sections. Slides were re-hydrated in 1X PBS and incubated in blocking solution containing 2% goat serum/0.01% Triton-X in 1X PBS for 1 hour at room temperature. Antibodies were dissolved in blocking solution and applied over night at 4˚C as follows: anti-PECAM1 (rat, 1:50; Thermo Fisher Scientific); anti-PH3 (rabbit, 1:1000; Millipore-Sigma) anti-Satb2 (mouse, 1:250; Abcam); anti-Tbr1 (rabbit, 1:250; Abcam). DAPI (1:4000; Millipore-Sigma) was used to detect cell nuclei. The following secondary antibodies were used: goat anti-rabbit CY3 (1:250; Jackson Immunoresearch); goat anti-mouse Alexa488 (1:1000; Jackson Immunoresearch); goat anti-rat CY3 (1:250; Jackson Immunoresearch).

## Image acquisition and cell density analyses

Image z-stacks were acquired using Zeiss LSM710 laser scanning confocal microscope and 20X objective. Obtained stacks spanned 12–14 µm and contained 5–7 optical slices. Brains were analyzed at the level of the presumptive visual cortex. ImageJ (NIH) software was used in manual and automated cell density analyses. Tbr1-expressing cells were manually counted in 60 x 60 µm regions of the lower cortical plate. Satb2-expressing cells were manually counted in 60 x 60 µm regions of the upper cortical plate. PECAM1-expressing blood vessels were manually counted in 100 x 100 µm regions of the upper cortical plate; fluorescence intensity and blood vessel areas were measured using ImageJ automated particle analysis module. PH3-expressing cells in the ventricular zone were manually counted on images collected at 10X magnification.

## Oxytocin measurements

Dams were administered probiotics in water, or water alone (control) starting at day 10.5 of pregnancy and plasma was collected 1 day after birth. Plasma oxytocin was extracted and measured according to the protocol supplied by the manufacturer of the ELISA kit (Enzo Life Sciences; ADI-900-153A-0001). Measurements were conducted using BioTek Synergy2 plate reader.

## Statistical analyses

Statistical analyses were performed with IBM SPSS Statistics (behavioral assays) and Graphpad (Prism) software. Data were checked for outliers using Grubbs test, normality using the Shapiro-Wilk test of normality and, for homogeneity of variances using Levene's test. Independent samples T-Test and to account for sex effects, two-way ANOVA analysis, with treatment (control and probiotic) and sex (male and female) as factors, was used to analyze the data. When there was a significant sex effect, a separate analysis by sex was conducted. If the data were not normally distributed, non-parametric Mann-Whitney test was used. In the fear conditioning experiment, repeated-measures ANOVA was also used to analyze the freezing behavior during the tests. Results are expressed as mean ± SEM unless otherwise specified.

# Results

## Maternal probiotic supplementation reduces anxiety-like offspring behavior in light-dark box test

To evaluate whether prenatal exposure to probiotics impacts anxiety-like behavior, mice were tested in the open field and light-dark box and novel object exploration test. In the light-dark box test, there was a significant interaction between the effects of sex and treatment on activity levels in the light zone ($F_{(1,36)} = 4.54$, p = 0.04, Fig 1A). A separate by sex t-test analysis showed that this effect was driven by female mice (Control 193.5 ± 45 cm and *L. lactis*

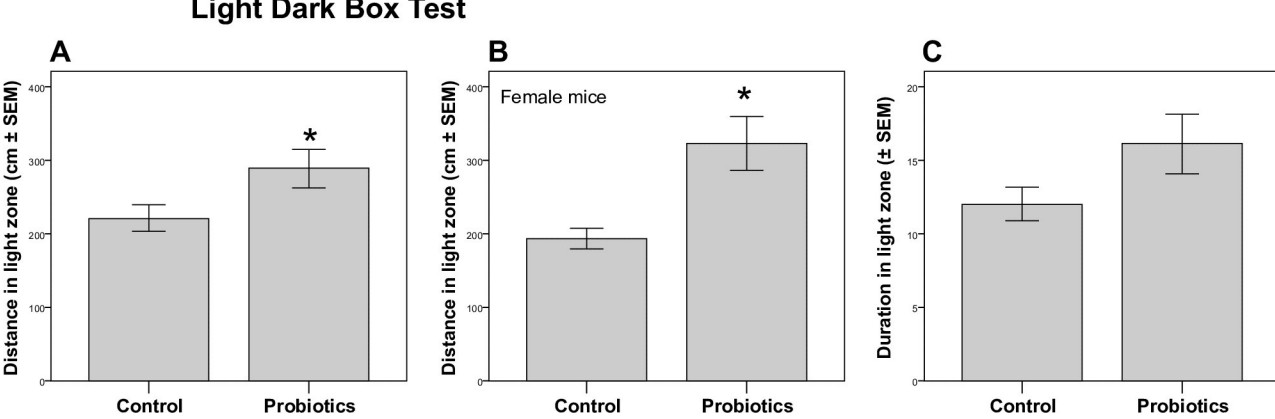

**Fig 1. Maternal probiotic supplementation reduces anxiety-like offspring behavior in light-dark box test.** (A) There was a significant interaction between the effects of sex and treatment on activity levels in the light zone (p = 0.04). (B) The difference in (A) was driven by female mice (p = 0.005). (C) Probiotic-exposed mice (male n = 9 and female n = 12) spent 10% more time than the controls ((male n = 9 and female n = 10) in the light zone, however this didn't reach significance.

323 ± 126 cm, p = 0.005, Student's t-test) while there was no difference in male mice (Control 258.2 ± 97 cm and *L. lactis* 252.5 ± 104 cm, p > 0.05, Student's t-test, Fig 1B). The *L. lactis*-exposed mice spent 10% more time in the light zone than the controls, but this difference did not reach significance with two-way ANOVA analysis for either simple main effects or interaction between these effects (p > 0.05, Fig 1C). Main effects analysis showed a trend for significance for the effects of in utero *L. lactis* exposure (F(1,36) = 3.817, p = 0.059), but there were no differences between males and females (p > 0.05).

In the open field, the latency to enter the center zone is lower in *L. lactis*-exposed mice than in control mice. However, this difference did not reach significance (Control 49.09 ± 15.7s and *L. lactis* 17.73 ± 3.9s, p = 0.06, S1A Fig) and there were no sex differences (p > 0.05). The activity levels were similar between the groups (Control 2547 ± 112 cm and *L. lactis* 2795 ± 103 cm, p > 0.05) and there were no sex differences (p > 0.05). We also looked at the percentage of time spent in the center as another measure of anxiety-like behavior. We did not find an effect of *L. lactis* treatment (Control 6.84 ± 1.63% and *L. lactis* 7.04 ± 1.38%, p > 0.05), but there was a strong main effect of sex, with female mice spending significantly more time in the center than male mice (F(1,37) = 100, p < 0.001, S1B Fig).

In the novel object exploration test, there was no difference in the latency to approach the object (Control 45.56 ± 9.53 s and *L. lactis* 29.98 ± 7.22 s, p > 0.05 S2A Fig), and there were no main effects of sex. Time spent investigating the object was not different between the groups (Control 8.94 ± 2.15 s and *L. lactis* 11.23 ± 2.86 s, p > 0.05 S2B Fig).

## Maternal probiotic supplementation differentially affects female mice in fear conditioning test

Treatment with probiotic did not affect conditioning, when analyzed with repeated measures ANOVA (p > 0.05). There was a significant interaction between sex and group (F(1.34) = 4.257, p = 0.047) with control female mice showing higher levels of freezing than males, however, the main effect of sex did not reach significance (p = 0.06; Fig 2A).

On the following day, mice were evaluated for their context-dependent learning. Mice were placed back into the original test chamber for a 5-min session, this time with no tone or foot shock. Probiotic treatment did not have an effect on the first minute response (p > 0.05) and there was no interaction between group and sex with univariate ANOVA (p > 0.05). However,

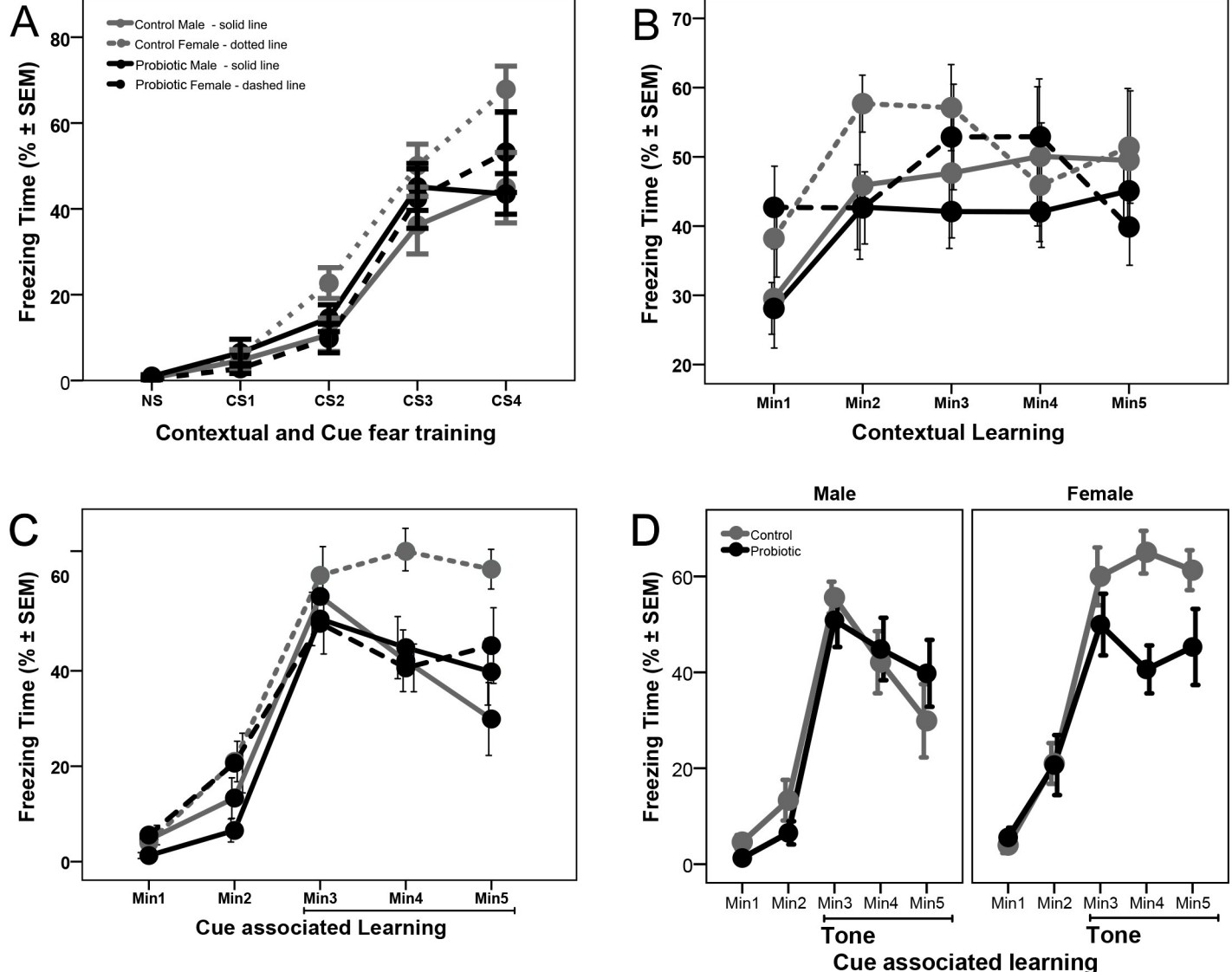

**Fig 2. Maternal probiotic supplementation differentially affects female mice in fear conditioning test.** (**A**) In utero exposure to probiotics did not affect conditioning of the mice. Control female mice (n = 10) showed higher levels of freezing than males (n = 8), however, the main effect of sex did not reach significance (p = 0.06). (**B**) Probiotic exposure did not have an effect on the contextual memory of mice. Though female mice (n = 10) showed higher levels of freezing than male mice (n = 10), separate sex analysis showed no treatment effect. (**C**) Probiotic exposure did not affect the first minute response to tone, but there is a significant treatment effect in female mice, shown in detail in panel (**D**).

there was a significant sex effect (F(1.34) = 4.287, p = 0.046) with female mice showing higher levels of freezing. Separate analysis in female and male mice showed no treatment effect (p > 0.05, S3A Fig). Repeated measures analysis, for the 5 minutes of the session, showed no differences between the groups and also no effect of sex (p > 0.05, Fig 2B). Analysis of average freezing time during the 5-min session showed no effect of treatment or sex (p > 0.05, S3B Fig).

On the third day of testing, associative learning to the tone cue was evaluated. The conditioning chambers were modified by turning off the white light and keeping only Near-Infrared light, by modifying the chamber using a black Plexiglas insert in an A-shape to change the wall and another insert to change the floor surface, and, by adding a novel odor (vanilla flavoring). Mice were placed in the modified chamber and allowed to explore it for a final 5-min session.

After 2-min, the acoustic stimulus was presented continuously for a 3-min period. Compared to control, probiotic treatment did not have an effect on the first minute response to tone ($p > 0.05$) and there was no interaction between group and sex, or a sex effect with univariate ANOVA ($p > 0.05$, S4A Fig). Repeated measures analysis for the 3 minutes of tone presentation showed no differences between the groups ($p > 0.05$) and neither the interaction between groups and sex, nor sex effect, reached statistical significance ($p > 0.05$, Fig 2C). However, separate analysis in female and male mice showed a significant treatment effect with repeated measures analysis for the 3 minutes of tone presentation in female mice ($F(1.18) = 5.700$, $p = 0.028$, Fig 2D) but not in male mice ($p > 0.05$). Treatment with probiotic reduced the freezing time in female mice in the second and third minutes of tone presentation, while the female mice in the control group maintained similar freezing levels. Overall, the above results suggest that probiotics are predominantly affecting female behavior after *in utero* exposure.

## *L. lactis* exposure during pregnancy induces an increase in maternal oxytocin levels

Oxytocin is a known hormonal mediator of maternal behavior [20]. To examine whether *L. lactis* exposure during pregnancy may increase maternal oxytocin levels, we measured oxytocin in plasma of treated and control dams 1 day after birth (Fig 3). We find that oxytocin levels are significantly increased in plasma of *L. lactis*-treated dams compared to control dams (Fig 3A; $p = 0.0008$ by Student's t-test). Oxytocin levels were not increased in plasma of pups at P28 (Fig 3B). These results suggest that probiotics exposure during pregnancy may modulate the levels of maternal oxytocin.

## Maternal supplementation with *L. lactis* modulates development of cortical vasculature in the pups

Maternal gut microbiota is known to influence blood vessel permeability in the developing embryo [21]. In this study, we examined whether the process of blood vessel formation during

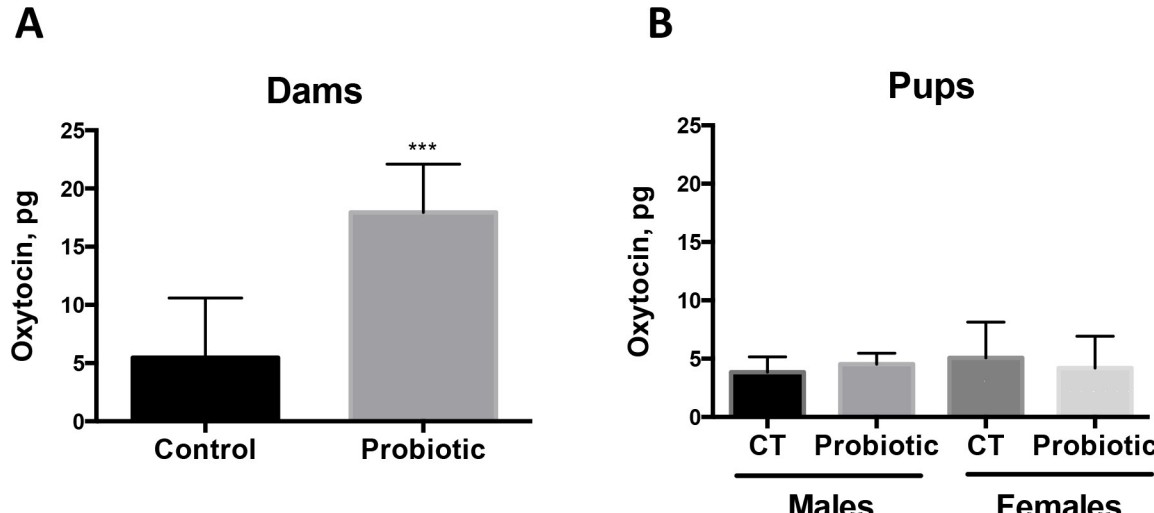

**Fig 3. Increase in plasma oxytocin levels in probiotics-treated dams.** (**A**) Oxytocin levels are increased in the plasma of dams treated with *L. lactis* between day 0.5 of pregnancy and 1 day after birth (n = 5) compared to Control dams (n = 8). P = 0.0008 by Student's t-test. (**B**) Plasma oxytocin levels remain unchanged in 3-month old male and female pups exposed to *L. lactis* between E0.5 and P1 (n = 5–6 per sex/group).

cortical development is impacted by maternal exposure to *L. lactis*. To this end, we evaluated the expression of PECAM1 (CD31), an endothelial cell marker, in the developing cortices of *L. lactis*-exposed compared to control P1 pups. We find that blood vessel number in the cortical plate is increased in probiotics-exposed male (p = 0.0007 by Student's t-test) and female (p = 0.0037 by Student's t-test) pups compared to control pups (Fig 4A–4C). Furthermore, expression levels of PECAM1, measured by fluorescence intensity, is increased in *L. lactis*-exposed pups (Fig 4B and 4E; p = 0.0002 by Student's t-test). Finally, analysis of blood vessel areas reveals an increase in the average blood vessel area in probiotics-exposed compared to control pups (Fig 4F; p = 0.0006). Together, these results demonstrate that maternal exposure to probiotics modulates formation of the vasculature in the developing embryonic brain.

### *In utero* exposure to *L. lactis* leads to structural changes in pyramidal neuronal cell layer organization of the cerebral cortex

Mouse cortical neurogenesis begins around embryonic day 11.5 (E11.5) and continues throughout gestation while cortical gliogenesis peaks during postnatal development [22]. Cortical pyramidal cell layers are generated in the inside-out manner, such that lower layer neurons (layers VI and V) are produced early in development, followed by the upper layer neurons (layers VI-II) [23]. To determine whether maternal supplementation with *L. lactis* from E10.5 through postnatal day 1 (P1) affects cellular composition of the developing cortex we examined the expression of the cortical lower layers marker, Tbr1 (layer VI), and upper layers marker, Satb2 (layers II-V), at P1. We find that the density of Tbr1-expressing neurons in *L. lactis*-exposed pups is increased (Fig 5A–5D; p < 0.0001 and p = 0.05, respectively, by Student's t-test). However, cortical layer VI thickness, as determined by the thickness of Tbr1-positive cell layer, is not statistically significant between groups in either male or female pups (Fig

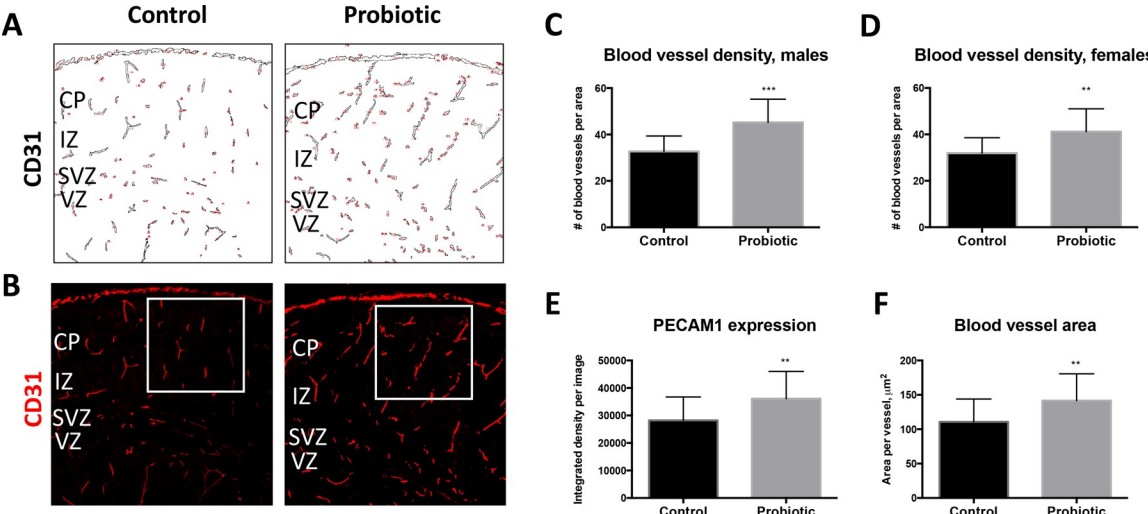

**Fig 4. Maternal exposure to probiotics modulates formation of cortical vasculature in the pups.** (**A**) Blood vessel outlines based on CD31 expression demonstrate increase in the density of cortical blood vessels in probiotic-exposed P1 pups compared to control pups. Counting masks are shown in red. (**B**) Immunofluorescent images of CD31 staining show increased staining intensity in P1 *L. lactis*-exposed compared to control cortices. White squares designate areas of blood vessel density analyses. (**C-D**) Increase in blood vessel density within the cortical wall is detected in both male (**C**; p = 0.0007 by Students t-test; control n = 18; *L. lactis* n = 9) and female (**D**; p = 0.0037 by Two-tailed Student's t-test; control n = 15; *L. lactis* n = 20) *L. lactis*-exposed versus control P1 pups. (**E**) Average expression levels of CD31 per blood vessel is increased in *L. lactis*-exposed P1 pup cortices (p = 0.0045 by Two-tailed Student's t-test; control n = 33; *L. lactis* n = 18). (**F**) Average blood vessel area is increased in P1 probiotic-exposed compared to control pups (p = 0.0045 by Two-tailed Student's t-test; control n = 33; *L. lactis* n = 18). Scale bar in (**B**): 80 μm. *CP*—cortical plate; *VZ*—ventricular zone.

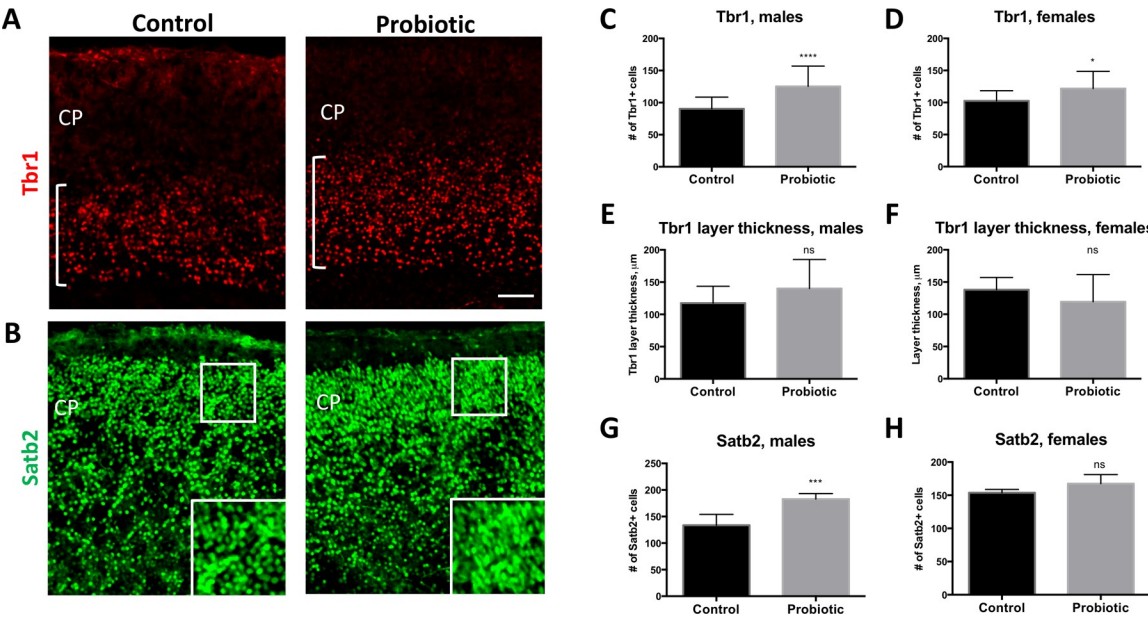

**Fig 5. Changes in cortical neuronal marker expression in P1 pups induced by maternal intake of probiotics during pregnancy.** (**A**) Expression of Tbr1 increased in the cortices of P1 pups exposed to *L. lactis*, compared to control pups. White brackets indicate Tbr1 layer thickness. (**B**) Density of Satb2-expressing cells in the upper layer of the cortical wall is increased in P1 *L. lactis*-exposed compared to control pups. Insets show higher magnification images of areas designated by white boxes. (**C-D**) Density of Tbr1-expressing cells is increased in male (**C**; p < 0.0001 by Two-tailed Student's t-test; control n = 26; *L. lactis* n = 13) and female (**D**; p = 0.05 by Two-tailed Student's t-test; control n = 10; *L. lactis* n = 18) probiotic-exposed P1 pups. (**E-F**) Tbr1 layer thickness was not significantly increased in probiotic-exposed compared to control P1 pups. (**G-H**) Density of Satb2-expressing cells is increased in the upper cortical layers (II-IV) of probiotic-exposed P1 male pups (**G**; p = 0.0006 by Two-tailed Student's t-test; control n = 7; *L. lactis* n = 5), but not female pups (**E**; p = 0.16 by Two-tailed Student's t-test; control n = 5; *L. lactis* n = 5) compared to control groups. Scale bar in (**A**): 80 μm. *CP*—cortical plate.

5A (bracket), 5E-5F). Similarly, the density of Satb2-expressing layer II-IV neurons is also increased, albeit only in male probiotics-exposed pups (Fig 5B and 5G; p = 0.0006 by Student's t-test) (Fig 5B and 5H).

Cortical pyramidal neurons and glia ultimately arise from neural progenitor cells (NPCs) that reside in the ventricular zone [22]. We therefore sought to examine whether the proliferative capacity of early postnatal NPCs may be affected by exposure to probiotic. We find that the density of mitotic NPCs expressing phosphorylated histone H3 (PH3), a marker of mitosis, is increased in the ventricular zone of probiotics-exposed female (Fig 6A and 6C; p = 0.0009 by Student's t-test), but not male pups, compared to P1 control female and male pups. Together, these results suggest that proliferative properties of postnatal cortical NPCs, as well as genesis and placement of cortical neurons within the cortical plate, are modulated by maternal exposure to probiotics.

## Discussion

As observed in the present study and in previous studies, prenatal probiotic treatment may have long-lasting effects on the cortex and cortical-driven behavior. Although the supposed sterility of the fetal gut is now disputed, previous studies have repeatedly shown that colonization of the neonatal microbiome can be altered via maternal stress and/or diet [24]. For example, the gut microbiomes of humans or rodents exposed to prenatal stress tend towards less diversity and numbers of lactic acid-producing strains compared to the microbiomes of unstressed offspring [2]. Additionally, developmental exposure to *Bifidobacterium* and

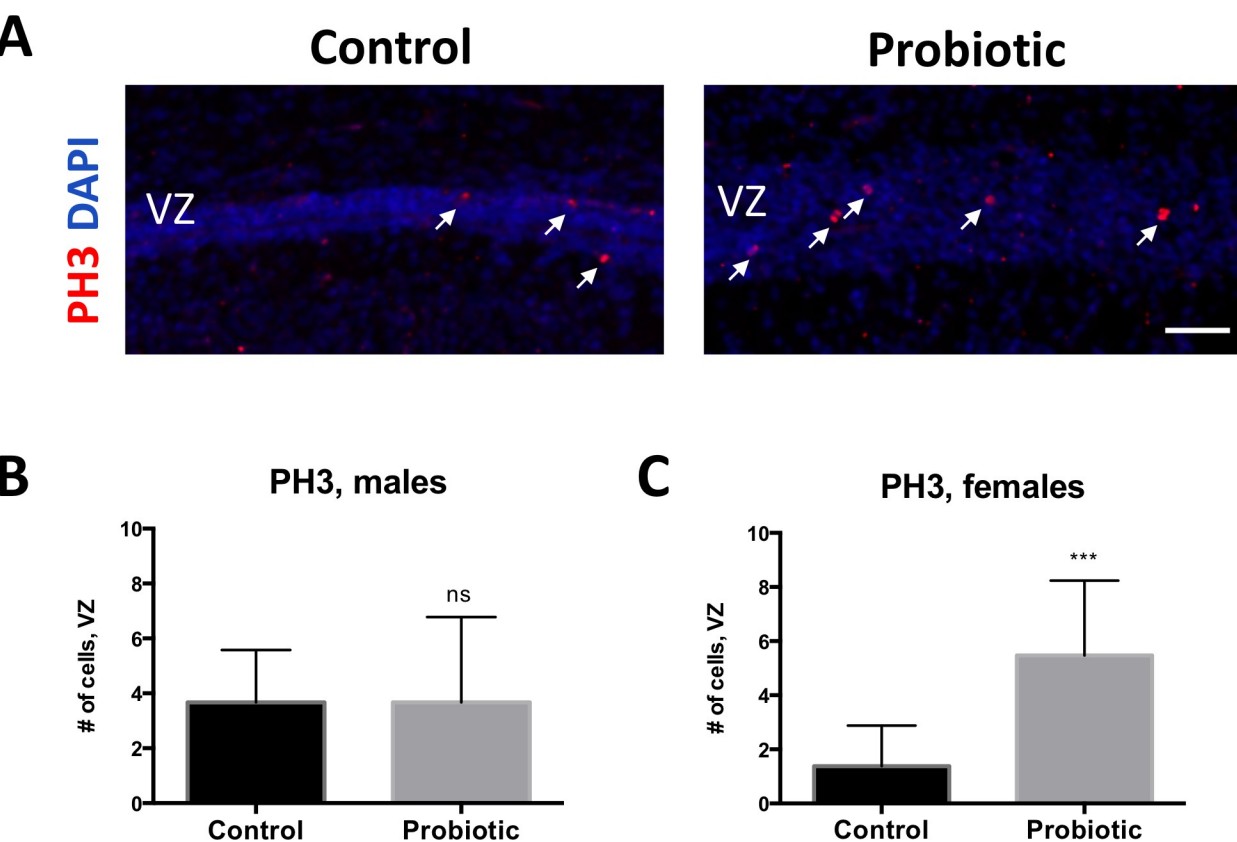

**Fig 6. Expression of mitotic marker PH3 is increased in probiotics-exposed female P1 pups.** (**A**) Larger numbers of PH3-expressing cells (white arrows) are observed in the cortical ventricular zone (VZ) of *L. lactis*-exposed versus control P1 pups. (**B-C**) Increase in PH3-expressing cells is significant in female pups (**C**; p = 0.0009 by Two-tailed Student's t-test; control n = 8; *L. lactis* n = 15), but not in male pups (**B**) (p > 0.9 by Two-tailed Student's t-test; control n = 18; *L. lactis* n = 15), compared to control groups. Scale bar in A: 60 μm.

*Lactococcus* strains have been shown to impact gut microbiome organization and immune responses, and subsequent stress reactivity in animal models [4, 25]. However, the above studies did not report the role of sex in modulation of stress via microflora interventions. Several groups have observed that maternal separation increased anxiety-like behaviors in the offspring, and probiotic treatment reversed stress effects in male rodents, but not female rodents [26, 27]. Emerging evidence also indicates that sex influences microbiome colonization and alters neurotransmitter metabolism into adulthood [28].

Anxiety in rodents can manifest through several types of behavior, such as open space avoidance or over-activity, and these symptoms vary according to age, sex and environmental factors [22]. Both females and males exhibited similar anxiety-like behaviors in the light dark box. However, probiotic treatment significantly increased light zone activity in the light-dark box in females only. In the fear conditioning test the control females' freezing was significantly higher than control males. This baseline sex effect is in keeping with previously reported results on differences between female and male mice in fear conditioning paradigms [29, 30]. Interestingly, a female-only effect of probiotics was observed in the cue associated fear conditioning, where probiotic-treated females exhibited significantly less freezing time after a fear-associated tone than the control females. Overall, these results may indicate that probiotics modulated anxiolytic behaviors that tend to be higher in females [31, 32].

It is unclear how probiotics are differentially affecting female anxiety-like behaviors. Maternal care may differ according to sex [33], and dams who received probiotics exhibited higher levels of plasma oxytocin, a reputed 'bonding' hormone. Oxytocin enhancement of nurturing behavior may increase female offspring resiliency even in the absence of overt prenatal stress, while additional nurturing for unstressed males may confer no additional benefit. Alternatively, it may be possible that the measurement parameters of fear conditioning and light-dark box are more sensitive to sex differences than the open field test, which is more prone to variability across testing environments and time periods [29, 31]. Further research is needed to understand how *in utero* probiotics exposure inhibits anxiety-like behaviors, in terms of maternal versus offspring stress responses, quality of maternal care, and offspring sex.

Changes in the expression patterns of cortical vasculature, neuronal and proliferation markers in probiotic-treated mice suggest that temporal progression of cortical layer development and vascularization may be modulated by the treatment. Appropriate development of cortical architecture is dependent on a signaling sequence of growth factors, vascular integration and cross-talk between cortical layer regions [34].In this study, P1 pup cortices were examined for changes in the cortical layer markers, Tbr1 and Satb2, a marker of mitosis, PH3, and a marker of angiogenesis, PECAM1 [35, 36]. Pronounced differences in cortical layer anatomy were observed in probiotics' treated mice. Specifically, probiotic treatment induced increases in layer VI and layers II-V neuronal marker expression. In addition, vascularization in the upper layers of the cerebral wall, measured by blood vessel density, as well as average blood vessel area, were also increased in response to probiotic exposure. Taken together these results suggest that patterning of cortical structural organization in the fetus may be sensitive to maternal dietary intake of probiotics.

The immune systems of probiotic-treated neonates may be better primed *in utero* to respond to pre- and post-birth microbial exposure compared to untreated pups. The developing brain is highly sensitive to maternal immune signaling, as indicated by disrupted cortical expression patterns after maternal infection [37, 38]. Kim et al recently reported that mouse dam injection of synthetic double-stranded RNA, mimicking viral infection, or gavage of segmented filamentous bacteria decreased cortical expression of Satb2 in offspring, and this effect was coupled with anxiogenic behavior in an open field and social approach test [39]. Furthermore, IL-17 production and gastrointestinal Th-17 cell activation was shown to be the mediator of Satb2 loss of expression. A marker of angiogenesis, CD31 (PECAM1), examined in the present study, also acts as a receptor for leukocyte activation and thus its upregulation in probiotic-treated brains may be part of an immune signaling cascade that links the gut to rate of cortical development [40]. An increase in CD31 expression may also underlie the observed increase of blood vessels after probiotic supplementation.

Radial glia populations that give rise to progenitors are mainly localized to regions adjacent to brain capillaries, and rates of proliferation are influenced by signaling factors from the periphery [34]. Along with the increased density of cortical upper layer blood vessels seen in probiotic brains, proliferation of cortical progenitors, as indicated by PH3, was also upregulated in female probiotic pups compared to control pups, and a trend effect was observed in male pups. In mice, cortical proliferation occurs during mid-gestation and subsides during early postnatal development—a temporal stage approximately equivalent to the middle of the second trimester in humans [34]. It is unknown if higher levels of PH3 expression during this time period, indicative of increased NPC proliferation and thus gliogenesis, is associated with improved resilience to stress. However, previous studies reported that decreased expression of PH3 in the developing cortex leads to anxiety-like behavior in male, but not female mice exposed to human antibodies [35]. Further studies are required to delineate sex-specific pathways regulating NPC proliferation and subsequent behavior.

In conclusion, we have shown that probiotic treatment during the latter half of gestation alters cortical cytoarchitecture in P1 offspring, suggesting that changes in cortical neuronal layer and vasculature protein expression at early postnatal stages may be associated with behavioral outcomes later in life [22–24]. Probiotic supplementation abrogated select anxiety-like behaviors, and improved emotional learning in females. It is presently unknown how observed changes in cortical cellular marker expression patterns may have contributed to altered behavior in probiotic-treated females, or whether changes in nurturing due to raised oxytocin levels may also have influenced sex-specific behavior differences. Further studies are needed to address the questions above and more clearly delineate the role of probiotics for human brain development.

## Supporting information

**S1 Data.**
(ZIP)

**S1 Fig.** Maternal probiotic supplementation (n = 22) did not have a significant effect on open field behavior, neither on the latency to enter the center zone compared to control mice (n = 18) (A) nor in the time spent in the center (B). We did observe a sex effect, where female mice (n = 22) spend significantly more time in the center than male mice (n = 18) (B).
(TIF)

**S2 Fig.** Maternal probiotic supplementation (n = 20) did not have a significant effect on novel object investigation, neither in the latency to approach the object compared to the control mice (n = 17) (A), nor in the time spent investigating the object (B).
(TIF)

**S3 Fig. Maternal probiotic supplementation did not have an effect in contextual learning.** (**A**) Female mice showed higher levels of freezing in the first minute of contextual learning but there was no treatment effect. (**B**) Probiotic exposure had no effect on average freezing time during the 5-min session showed no effect of treatment or sex.
(TIF)

**S4 Fig. Cue associated learning.** Probiotic exposure did not have an effect on the first minute response to tone in either male (control n = 8 and probiotic n = 10) or female mice (control n = 10 and probiotic n = 10).
(TIF)

## Acknowledgments

We thank Dr. Steven Zeisel for his input in the design of this study; Dr. David Horita for critically evaluating this manuscript; and Dr. Steven Oreña for technical assistance with oxytocin assays.

## Author Contributions

**Conceptualization:** Jonas Hauser, Ellen Siobhan Mitchell.

**Data curation:** Eneda Pjetri.

**Formal analysis:** Natalia Surzenko, Carolyn A. Munson.

**Funding acquisition:** Natalia Surzenko.

**Investigation:** Natalia Surzenko, Eneda Pjetri, Carolyn A. Munson, Walter B. Friday.

**Methodology:** Natalia Surzenko, Eneda Pjetri, Carolyn A. Munson, Jonas Hauser.

**Project administration:** Natalia Surzenko, Eneda Pjetri, Ellen Siobhan Mitchell.

**Resources:** Natalia Surzenko.

**Software:** Natalia Surzenko, Eneda Pjetri.

**Supervision:** Eneda Pjetri.

**Validation:** Eneda Pjetri.

**Visualization:** Natalia Surzenko, Carolyn A. Munson, Walter B. Friday.

**Writing – original draft:** Natalia Surzenko, Eneda Pjetri, Jonas Hauser, Ellen Siobhan Mitchell.

**Writing – review & editing:** Natalia Surzenko, Eneda Pjetri, Jonas Hauser, Ellen Siobhan Mitchell.

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
