## [Decision Letter · Decision Letter 0]

28 Oct 2019

PONE-D-19-25682

Prenatal exposure to the probiotic Lactococcus lactis decreases anxiety-like behavior and modulates cortical cytoarchitecture in a sex specific manner.

PLOS ONE

Dear Dr Hauser,

Thank you for submitting your manuscript to PLOS ONE. After careful consideration, we feel that it has merit but does not fully meet PLOS ONE’s publication criteria as it currently stands. Therefore, we invite you to submit a revised version of the manuscript that addresses the points raised during the review process. Please find the reviewer comments below.

We would appreciate receiving your revised manuscript by Dec 12 2019 11:59PM. To enhance the reproducibility of your results, we recommend that if applicable you deposit your laboratory protocols in protocols.io, where a protocol can be assigned its own identifier (DOI) such that it can be cited independently in the future. For instructions see: http://journals.plos.org/plosone/s/submission-guidelines#loc-laboratory-protocols

We look forward to receiving your revised manuscript.

Kind regards,

Judith Homberg

Academic Editor

PLOS ONE

Journal Requirements:

1. 

2.  During our initial internal evaluation of your submission, we noted that your study reports experiments carried out in a mouse model. Therefore, at this time, we ask that you please update your title to reflect the experimental animal model. For example, Prenatal exposure to the probiotic Lactococcus lactis decreases anxiety-like behavior and modulates cortical cytoarchitecture in a sex specific manner in mice.

"I have read the journal's policy and the authors of this manuscript have the following competing interests: ESM and JH are or were employees of Société des Produits Nestlé SA, a for-profit institution, and received support in the form of salaries. "

Reviewers' comments:

Reviewer's Responses to Questions

**Comments to the Author**

1. Is the manuscript technically sound, and do the data support the conclusions?

Reviewer #1: Yes

2. Has the statistical analysis been performed appropriately and rigorously? 

Reviewer #1: No

3. Have the authors made all data underlying the findings in their manuscript fully available?

Reviewer #1: Yes

4. Is the manuscript presented in an intelligible fashion and written in standard English?

Reviewer #1: Yes

5. Review Comments to the Author

Reviewer #1: Manuscript No. PONE-D-19-25682

In this work the Authors treated pregnant dams with the probiotic Lactococcus lactis. Then, behavioral, biochemical and histological experiments were carried out on the offspring or their mothers. Findings suggest that exposure dams to the probiotic may exert sex-specific long-term effects on cortical development and anxiety related behavior in the offspring.

Gut microbiota and probiotic bacteria are widely in focus of many researches. This work also adds potential interesting results to attempts of others. However, the manuscript (MS) to be matured needs a substantial revision.

My comments:

1-It seems the main aim of the study was not to examine sex dependency of probiotic treatment, so, the Authors need to also focus on the main goal.

2-In Abstract, line 34, Authors say "probiotics treatment throughout pregnancy". But according to Methods the treatment occurred during second half of pregnancy.

3-While not mothers and not offspring were exposed to stress, a question raise why the Authors introduced the pups to tests that evaluate anxiety stress.

4-As far as I know Bifidus (as mentioned in this MS) is not a popular name to replace Bifidobacterium species.

5-Some sentences refer to work of others but no reference is given. For instance, line 56-58 and line 60-61.

6-I could not understand the dose of probiotic. Further, the Lactobacillus is an anaerobic bacterium. Diluting probiotic in water leads to abrupt decrease of dosage. Therefore, the Authors need to explain how they kept the dosage consistent.

7-In Results Authors must give degree of freedom (F…,…=..) when using ANOVA as they did for light-dark box test. But in some other cases they do not mention F. Moreover, to show a significant difference it is not enough to give just F, but the P values from post test is necessary.

8- Not necessary to mention (SEM) in Y axis of figures (like fig 2).

9- According to “Statistical analyses” error bars indicate SEM. Based on, some comparisons like those in the curves E and F in figures 4 and 5 or curve B in figure 6 seem not to be significant. Please check them.

10-It is not clear how the Authors compared different groups, especially male and female groups. For instance, in many cases it is stated that there was a significant difference between females and males. Question is, in which group, probiotic treated or control group. A comparison of male and female animals in the control group is not valuable, unless the difference is evident between those treated with probiotics. Please clear it.

11-The real Discussion begins from line 383. But from this part on the findings are discussed with just 3 references. Thus, the Authors are expected to present a stronger discussion which also cover all findings.

12-Despite the MS is almost well written, however, concerning English language, it requires a whole revision.

13-Please follow consistency over the MS. For instance, Lactobacillus lactis is mentioned differently in lines 73 and 77.

14-Line 91, what do you mean: (John and Bell 1976, Council 1995).

6. PLOS authors have the option to publish the peer review history of their article (what does this mean?). If published, this will include your full peer review and any attached files.

Reviewer #1: Yes: Mahmoud Salami

---

## [Author Response · Author response to Decision Letter 0]

14 Jan 2020

Dear Prof. Homberg,

We are delighted to hear that the Reviewers found our manuscript technically sound, with data supporting the conclusions. We thank the Reviewers for their time and insightful comments. Below, we provide the details of our manuscript revisions, which reflect the Reviewer’s comments and concerns.

Reviewer comment 1:

1-It seems the main aim of the study was not to examine sex dependency of probiotic treatment, so, the Authors need to also focus on the main goal.

Author response:

In agreement with the Reviewer’s comment, we have ensured that the text referring to sex-dependent variation in responses to probiotic treatment does not appear in the Introduction section of the manuscript, but is included in the Discussion (pages 17-18)

Reviewer comment 2:

2-In Abstract, line 34, Authors say "probiotics treatment throughout pregnancy". But according to Methods the treatment occurred during second half of pregnancy.

Author response:

We have corrected this statement in the manuscript Abstract to more accurately reflect the period of probiotic treatment during the second half of pregnancy (line 30).

Reviewer comment 3:

3-While not mothers and not offspring were exposed to stress, a question raise why the Authors introduced the pups to tests that evaluate anxiety stress.

Author response:

Based on the existing literature, linking treatment with probiotic and reduced anxiety (reviewed in Lach et al. Neurotherapeutics 15(1):36-59 2018), we hypothesized that our treatment might result in a decrease of anxiety in the mice. Considering that the behavior in the light-dark box test and in the open field not only is sensible to manipulation increasing anxiety, but also reflect an important variability due to innate higher or lower anxiety levels, we expected these tests to detect an anxiolytic effect of our treatment.

Reviewer comment 4:

4-As far as I know Bifidus (as mentioned in this MS) is not a popular name to replace Bifidobacterium species.

Author response:

We have corrected the text referring to the Bifidobacterium species throughout the manuscript.

Reviewer comment 5:

5-Some sentences refer to work of others but no reference is given. For instance, line 56-58 and line 60-61.

Author response:

We have added the appropriate references to the statements included in lines 56-58 and 60-61, and have checked the remainder of the manuscript for proper use of citations.

Reviewer comment 6:

6-I could not understand the dose of probiotic. Further, the Lactobacillus is an anaerobic bacterium. Diluting probiotic in water leads to abrupt decrease of dosage. Therefore, the Authors need to explain how they kept the dosage consistent.

Author response:

In the Materials and Methods section of the manuscript, we mention the dose of the probiotic diluted daily in the drinking water based on the concentration of the powder, which was determined at the beginning and confirmed at the end of the study.

In reference to the probiotic stability in water and the actual dose received by animals - the Lactobacillus species are facultative aerobes, so they can tolerate oxygen very well. We therefore did not expect a drastic fast reduction of probiotic concentration in water due to exposure to oxygen.

Reviewer comment 7:

7-In Results Authors must give degree of freedom (F…,…=..) when using ANOVA as they did for light-dark box test. But in some other cases they do not mention F. Moreover, to show a significant difference it is not enough to give just F, but the P values from post test is necessary.

Author response:

Univariate (two-way) ANOVA analysis, with treatment (control and probiotic) and sex (male and female) as factors, was used to analyze some of the data. Both factors have only two groups and no post-hoc analysis is available. For our fear conditioning experiments we ran the repeated measures ANOVA. In the results section we have made sure all the significant results are shown with their p-value and F-value when univariate or repeated measures ANOVA was used for analysis. 

Reviewer comment 8:

8- Not necessary to mention (SEM) in Y axis of figures (like fig 2).

Author response:

Some of our figures are shown as means with standard error of means (SEM). For these figures, we would like to keep the SEM in Y-axis as it helps the readers to understand the data distribution. The error bars may also represent standard deviation and we want to make the correct annotation. 

Reviewer comment 9:

9- According to “Statistical analyses” error bars indicate SEM. Based on, some comparisons like those in the curves E and F in figures 4 and 5 or curve B in figure 6 seem not to be significant. Please check them.

Author response:

To address the Reviewer’s concern with regards to statistical significance of our findings, we have confirmed the analyses and the P values reported in Figures 4, 5 and 6. In addition, the original data is available at ResearchGate for verification.

Reviewer comment 10:

10-It is not clear how the Authors compared different groups, especially male and female groups. For instance, in many cases it is stated that there was a significant difference between females and males. Question is, in which group, probiotic treated or control group. A comparison of male and female animals in the control group is not valuable, unless the difference is evident between those treated with probiotics. Please clear it.

Author response:

The statistical approach for the analyses was to use a two-way ANOVA with treatment (control and probiotic) and sex (male and female) as factors, this provided us with the statistical mean to identify: main effect of sex (males different from females independently of the treatment), main effect of treatment (control different from probiotic independently of sex) or an interaction of the two (effect of the treatment is dependent of the sex). Thus, when we report a main effect of sex in the manuscript, we report a difference between the males and the females, independently of the treatment effect (e.g. the increased levels of freezing observed in females during the 2nd day of fear conditioning). With this model we also can better understand if any treatment effect is dependent on the sex of the subject (e.g. the difference observed in freezing time during contextual and cue fear training, where the probiotic effect was observed in females only, visible as the decrease of freezing time in treated females compared to control females, while such an effect was not observed in males). We have included the information on which groups were compared in the manuscript text by stating whether the statistical difference referred to a main effect of gender or an interaction of treatment and gender (see examples aforementioned).

Reviewer comment 11:

11-The real Discussion begins from line 383. But from this part on the findings are discussed with just 3 references. Thus, the Authors are expected to present a stronger discussion which also cover all findings.

Author response:

To address this concern, we have revised the Discussion section of the manuscript, which now addresses all the findings and includes appropriate references.

Reviewer comment 12:

12-Despite the MS is almost well written, however, concerning English language, it requires a whole revision.

Author response:

To ensure the manuscript is written in proper English, we have enlisted the help of a native English-speaking scientist in evaluating the manuscript text and suggesting revisions where necessary.

Reviewer comment 13:

13-Please follow consistency over the MS. For instance, Lactobacillus lactis is mentioned differently in lines 73 and 77.

Author response:

We have ensured the consistency of the text referring to Lactococcus lactis species used in this study.

Reviewer comment 14:

14-Line 91, what do you mean: (John and Bell 1976, Council 1995).

Author response:

We have ensured that all references cited in the text follow a consistent format.

---

## [Decision Letter · Decision Letter 1]

18 Feb 2020

PONE-D-19-25682R1

Prenatal exposure to the probiotic Lactococcus lactis decreases anxiety-like behavior and modulates cortical cytoarchitecture in a sex specific manner.

PLOS ONE

Dear Dr Hauser,

Thank you for submitting your manuscript to PLOS ONE. After careful consideration, we feel that it has merit but does not fully meet PLOS ONE’s publication criteria as it currently stands. Therefore, we invite you to submit a revised version of the manuscript that addresses the points raised during the review process. Please find the reviewer comments below.

We would appreciate receiving your revised manuscript by Apr 03 2020 11:59PM. To enhance the reproducibility of your results, we recommend that if applicable you deposit your laboratory protocols in protocols.io, where a protocol can be assigned its own identifier (DOI) such that it can be cited independently in the future. For instructions see: http://journals.plos.org/plosone/s/submission-guidelines#loc-laboratory-protocols

We look forward to receiving your revised manuscript.

Kind regards,

Judith Homberg

Academic Editor

PLOS ONE

Reviewers' comments:

Reviewer's Responses to Questions

**Comments to the Author**

1. If the authors have adequately addressed your comments raised in a previous round of review and you feel that this manuscript is now acceptable for publication, you may indicate that here to bypass the “Comments to the Author” section, enter your conflict of interest statement in the “Confidential to Editor” section, and submit your "Accept" recommendation.

Reviewer #1: (No Response)

2. Is the manuscript technically sound, and do the data support the conclusions?

Reviewer #1: Yes

3. Has the statistical analysis been performed appropriately and rigorously? 

Reviewer #1: No

4. Have the authors made all data underlying the findings in their manuscript fully available?

Reviewer #1: Yes

5. Is the manuscript presented in an intelligible fashion and written in standard English?

Reviewer #1: Yes

6. Review Comments to the Author

Reviewer #1: 1-The 3rd question was:

"While not mothers and not offspring were exposed to stress, a question raise why the Authors introduced the pups to tests that evaluate anxiety stress." The Authors say "we hypothesized that our treatment might result in a decrease of anxiety in the mice." However, they research shows that there is no difference between control and probiotic treated animals when testing anxiety.

2- the 7th question was:

"In Results Authors must give degree of freedom (F…,…=..) when using ANOVA as they did for light-dark box test. But in some other cases they do not mention F. Moreover, to show a significant difference it is not enough to give just F, but the P values from post test is necessary."

The Authors say "Univariate (two-way) ANOVA analysis, with treatment (control and probiotic) and sex (male and female) as factors, was used to analyze some of the data. Both factors have only two groups and no post-hoc analysis is available." However, suppose you compare only two lines of data, therefore, ANOVA is not a right analysis; in this case an analysis like t-test is more appropriate.

3-Also: When the between group differences are not statistically significant it means there is no any difference. So, a title like (Maternal probiotic supplementation reduces anxiety-like offspring behavior in light-dark box test) is not acceptable.

7. PLOS authors have the option to publish the peer review history of their article (what does this mean?). If published, this will include your full peer review and any attached files.

Reviewer #1: No

---

## [Author Response · Author response to Decision Letter 1]

3 Apr 2020

Reviewer’s comment 1:

The 3rd question was:

"While not mothers and not offspring were exposed to stress, a question raise why the Authors introduced the pups to tests that evaluate anxiety stress." The Authors say "we hypothesized that our treatment might result in a decrease of anxiety in the mice." However, they research shows that there is no difference between control and probiotic treated animals when testing anxiety.

Author’s response:

We thank the Reviewer for pointing to the potential confusion in the manuscript. Our initial response to the Reviewer was: “Based on the existing literature, linking treatment with probiotic and reduced anxiety (reviewed in Lach et al. Neurotherapeutics 15(1):36-59 2018), we hypothesized that our treatment might result in a decrease of anxiety in mice. Considering that the behavior in the light-dark box test and in the open field not only is sensible to manipulation increasing anxiety, but also reflect an important variability due to innate higher or lower anxiety levels, we expected these tests to detect an anxiolytic effect of our treatment.” 

Furthermore, in the Introduction section of our manuscript (pp.3-4) we have emphasized that experimental manipulations affecting the microbiota, such as probiotic treatment or absence of the microbiota, have shown to impact anxiety behaviors in multiple models. Thus, we appear to have addressed the reasons for testing the link between the probiotic treatment and anxiety.

With respect to the “difference between control and probiotic-treated animals when testing anxiety”, we have revised the Results section of our manuscript, titled “Maternal probiotic supplementation reduces anxiety-like offspring behavior in light-dark box test” (pp. 9-11), and a corresponding Figure 1, along with its legend, to highlight the significant findings.

Specifically, we replaced panel B in the revised Figure 1 with a graph demonstrating the significant impact of probiotic treatment on the distance moved in the light zone in female mice. In summary, revised Figure 1 now demonstrates (A) statistically significant effect of probiotic treatment on the distance moved in the light zone when mice of both sexes are considered in the analyses, (B) statistically significant effect of probiotic treatment on the distance moved in the light zone in female mice, which appear to drive the overall effect, and (C) a trend towards significance for the time spent in the light zone when both sexes are considered (p = 0.059), with probiotic-treated animals spending 10% more time in the light zone. Together with the trend towards significance in the effect of the probiotic treatment on the latency to enter the center zone in the open field test (Figure S1; p = 0.06), our data strongly support an anxiolytic effect of the probiotic treatment. 

Reviewer’s comment 2-1:

2- the 7th question was:

"In Results Authors must give degree of freedom (F…,…=..) when using ANOVA as they did for light-dark box test. But in some other cases they do not mention F. Moreover, to show a significant difference it is not enough to give just F, but the P values from post test is necessary."

Author’s response:

To address this concern, we have ensured to include information on both - the degrees of freedom and the p values - in the revised manuscript text and figure legends, where appropriate.

Reviewer’s comment 2-2:

The Authors say "Univariate (two-way) ANOVA analysis, with treatment (control and probiotic) and sex (male and female) as factors, was used to analyze some of the data. Both factors have only two groups and no post-hoc analysis is available." However, suppose you compare only two lines of data, therefore, ANOVA is not a right analysis; in this case an analysis like t-test is more appropriate.

Author’s response:

We thank the Reviewer for their careful consideration of the statistical analyses performed in our manuscript. We agree with the Reviewer that when comparing only two lines of data, a t-test would be more appropriate. However, considering that in addition to the effect of the treatment we also had to consider sex, we were conservative with our data analyses and utilized a method allowing to assess all the important factors, as well as the interactions between these factors. Therefore, ANOVA is an appropriate method of data analysis when multiple factors are taken into consideration.

Importantly, while Reviewer’s comment “suppose you compare only two lines of data, therefore, ANOVA is not a right analyses; in this case an analysis like a t-test is more appropriate” is technically correct, there is no difference in the results generated by a t-test and an ANOVA when comparing only two groups - presenting the data from an ANOVA analysis is equivalent to presenting the data from a t-test. Therefore, we feel that to keep the manuscript consistent with regards to data analysis methods, presenting the ANOVA analyses throughout the manuscript is appropriate.

Reviewer’s comment 3:

3-Also: When the between group differences are not statistically significant it means there is no any difference. So, a title like (Maternal probiotic supplementation reduces anxiety-like offspring behavior in light-dark box test) is not acceptable.

Author response:

In our answer to Reviewer’s comment 1 above, we have summarized the significant results of our study in relation to the anxiolytic effects of the probiotic treatment in light-dark box test. Briefly, we have found a significant effect of the probiotic treatment on the distance moved in the light zone (Figure 1A), which is driven by female mice (Figure 1B), and a trend towards significance for the effect of the probiotic treatment on the duration of time spent in the light zone (p = 0.059; Figure 1C). Together, these results justify the title of this manuscript section.

---

## [Decision Letter · Decision Letter 2]

28 Apr 2020

Prenatal exposure to the probiotic Lactococcus lactis decreases anxiety-like behavior and modulates cortical cytoarchitecture in a sex specific manner.

PONE-D-19-25682R2

Dear Dr. Hauser,

We are pleased to inform you that your manuscript has been judged scientifically suitable for publication and will be formally accepted for publication once it complies with all outstanding technical requirements.

With kind regards,

Judith Homberg

Academic Editor

PLOS ONE

Additional Editor Comments (optional):

Reviewers' comments:

Reviewer's Responses to Questions

**Comments to the Author**

1. If the authors have adequately addressed your comments raised in a previous round of review and you feel that this manuscript is now acceptable for publication, you may indicate that here to bypass the “Comments to the Author” section, enter your conflict of interest statement in the “Confidential to Editor” section, and submit your "Accept" recommendation.

Reviewer #1: (No Response)

2. Is the manuscript technically sound, and do the data support the conclusions?

Reviewer #1: Yes

3. Has the statistical analysis been performed appropriately and rigorously? 

Reviewer #1: N/A

4. Have the authors made all data underlying the findings in their manuscript fully available?

Reviewer #1: Yes

5. Is the manuscript presented in an intelligible fashion and written in standard English?

Reviewer #1: Yes

6. Review Comments to the Author

Reviewer #1: (No Response)

7. PLOS authors have the option to publish the peer review history of their article (what does this mean?). If published, this will include your full peer review and any attached files.

Reviewer #1: Yes: Mahmoud Salami

---

## [Editor Report · Acceptance letter]

29 Jun 2020

PONE-D-19-25682R2 

Prenatal exposure to the probiotic Lactococcus lactis decreases anxiety-like behavior and modulates cortical cytoarchitecture in a sex specific manner. 

Dear Dr. Hauser:

I'm pleased to inform you that your manuscript has been deemed suitable for publication in PLOS ONE. Congratulations! Your manuscript is now with our production department. 

Kind regards, 

on behalf of

Dr. Judith Homberg 

Academic Editor

PLOS ONE